# Evaluation of Plan Implementation in the Fast-Growing Chinese Mega-City: A Case of a Polycentric System in Hangzhou Core Built-Up Area

**Liang Ding [1], Cheng Shi [2,*] and Xinyi Niu [2]**

[1] College of Civil Engineering and Architecture, Zhejiang University of Technology, Hangzhou 310023, China; liangd05813@zjut.edu.cn

[2] College of Architecture and Urban Planning, Tongji University, Shanghai 200092, China; niuxinyi@tongji.edu.cn

**\*** Correspondence: chengshi@tongji.edu.cn

**Abstract:** Previous evaluations of plan implementation focused on whether the materiality construction was in accordance with the plan. Without proper data, it is difficult to confirm whether the planning goals have been achieved. In this study, two types of big data have been used—full sample built-environment data and mobile-positioning big data—to evaluate the results of the implementation of the polycentric system in master planning in the Hangzhou core built-up area. Using the full sample built-environment data, the evaluation of materiality construction will be more objective and accurate. Using the mobile-positioning big data, the evaluation of the planning goals can be realized; this was almost impossible in the past. However, two aspects are considered: whether daily public activities, such as employment and recreation, have been dispersed from the old city and subsequently re-gathered in multiple centers outside the old city, and whether the polycentric system aids in optimizing the spatial relationship between residence and public activity. The following conclusions were drawn. In terms of actual materiality construction, the results showed minimal discrepancy from the plan. Fifteen city-level public centers have been constructed at principal, secondary, and sub-secondary levels. However, the polycentric system failed to achieve the expected effects of the planning goals. First, the public centers contributed in the gathering of public activities; however, the level of gathering at the newly built-up public centers was considerably lower than that at traditional public centers. Second, the public centers failed to encourage people to visit the nearest blocks for daily public activities, mainly because of the path dependence on the traditional centers in the process of multi-centralization and over-rapid expansion of the city. Owing to this, residents did not have sufficient time to adjust to the spatial relationship between the residence and daily public activities.

**Keywords:** evaluation of plan implementation; polycentric system; mega-city; big data

## 1. Introduction

The emergence of polycentric systems is an irresistible trend of the urban growth and service industry spillover caused by the excessive agglomeration of elements in an original center, which can theoretically moderate the problem of agglomeration diseconomy caused by a single center [1,2]. A polycentric system has two objectives: firstly, it aims to alleviate the excessive concentration of daily public activities in old cities by dispersing and re-gathering them in multiple peripheral centers; secondly, it aims to optimize the spatial relationship between residence and daily public activities, providing daily public services for residents living outside the old city, thus reducing the length of the travel distance. The polycentric system can moderate "big city malaises" only when the two objectives

are realized simultaneously. If pubic activities are not dispersed and re-gathered, even if multiple centers were built in peripheral areas they would not function. On the other hand, if public activities were dispersed and re-gathered but residents of peripheral areas would still need to travel to the old city for their daily activities and residents of the old city to peripheral centers, the polycentric system would contradict its original planning objective. Taking the influence of polycentric system on commuting as an example, several research findings have shown that the polycentric system is conducive to reducing commuting time and distance [3–5]. However, this may also lead to an increase in commuting time and distance [6–9], which are related to the size of the city and the implementation time of the polycentric system [10,11]. Therefore, it is necessary to evaluate the implementation of the polycentric system in a timely fashion to identify problems and make timely adjustments in relevant policies.

In China, the polycentric system is one of the main tasks at the stage of a master plan. Its major tasks are to arrange the location of public centers and to determine the level of each center to identify the gathering location, and the scale of the public service facilities (business, commerce, culture, etc.) of the future urban space. The local government hopes to guide the construction through a plan, and to finally achieve the aforementioned two objectives of the polycentric system by building a polycentric material space. The implementation period of the polycentric system is consistent with the master planning, i.e., approximately 20 years. Implementation of the last round of master planning (from 2000 to 2020 for most cities) coincides with the accelerating expansion of urban space in China. According to statistics from the Ministry of Housing and Urban-Rural Development of the People's Republic of China (MOHURD), urban built-up areas increased from 22,000 km$^2$ in 2000 to 56,000 km$^2$ in 2017, which is an increase of 1.5 times in 18 years. The urban spatial structure has been evolving from the single center system (that used to be limited to the old city) to a polycentric system. At present, the final round of master planning has entered the final implementation period, and it is the appropriate time to evaluate the implementation results of the then-defined polycentric system.

Although evaluation related to the polycentric system has been practiced for several years, evaluations mainly focused on whether the materiality construction has been realized. Without proper data, it is difficult to confirm whether the planning goals have been achieved. With data availability and technical progress in recent years, new ideas and methods have been brought to the evaluation. On the one hand, the full sample built-environment data are helpful in analyzing construction of the material space of the entire city more objectively and accurately, and in improving the evaluation of material construction. On the other hand, mobile-positioning big data enables us to understand the real public activity characteristics of residents, to realize evaluation of the planning objectives—which could not have been completed in the past—and this is the focus of the presented study.

The remainder of the paper is structured as follows. In Section 2, the literature related to plan implementation evaluation and mainly research data will be discussed. In Section 3, the evaluation framework for the plan implementation of the polycentric system will be constructed. In Section 4, the implementation results will be evaluated by using the polycentric system in the master planning at the Hangzhou core built-up area as a case. Finally, a discussion and the summary of the research conclusions will be presented.

## 2. Literature Review

### 2.1. Research of Plan Implementation Evaluation

Conformance is one of the criteria for the evaluation of plan implementation [12–17]. Although currently, it is widely believed that because planning is uncertain, implementation results cannot be expected to fully conform to the plan [18]. To evaluate whether a plan implementation is successful or not, the performance should be considered as well. In other words, as long as the plan is regarded as a part of the decision-making process and the implementation results are beneficial, then plan implementation can be considered successful [12,15,17]. However, because a plan differs from other policies and planning results are typically expressed as blueprints, conformance evaluation is still

necessary. Studies made progress first regarding evaluating materiality construction conformance: Alterman et al. [19] compared the conformance degree between an outline plan, a detailed plan, and a permit using the grid over-layers method, and analyzed the reasons influencing plan implementation from three aspects, namely, political factors, attributes of the plan, and the urban system. Since 2000, as the data conditions improved, the number of research results has been gradually increasing. For instance, Berke et al. [20], using low-impact design techniques as an example, compared the conformance degree between a local plan and a permit, and analyzed the reasons that caused the implementation to deviate from the plan in terms of plan quality, enforcement style, and agency staff capacity. Brody et al. [21], using wetland development as a case, compared the conformance degree between the actual development and the plan, and explained the cause of non-conformance in terms of geography, social demographics, and policies and markets. Tian et al. [22] compared data on land use and actual land development of the master plan, calculated the proportions of accordance, deviation, and unfulfillment of all types of land, and analyzed the influence of political–institutional factors, attributes of the plan, and urban system factors on plan implementation. Loh [17] compared the differences between planned and actual land development, and divided non-conformance into three types: undeveloped, temporary development, and development not conforming to the plan, proposing that focus should be placed on the development not conforming to the plan.

According to the definition of conformance by Alexander et al. [12], whether the blueprint is implemented is only one aspect of conformance assessment, and whether plan implementation effect meets expectations also needs to be analyzed. More specifically, regardless of whether the outcome of actual materiality construction is not in line with the blueprint, if implementation results are consistent with the original plan objectives, the plan is still considered as having been successfully implemented [13]. Talen [13] evaluated the plan of distribution of public facilities. However, he did not compare whether the spatial location of the plan and the facility construction are consistent with one another; instead, he simulated whether the spatial relationship between public facilities and residents after plan implementation is similar to what the original plan envisaged. Zhong et al. [23] conducted an implementation evaluation of the use plan of state land, in which they did not employ actual land use as a comparison; instead, they selected the three goals that the plan is intended to achieve—farmland protection, the control of the expansion of construction land, and ecological protection—as comparison objects. In their evaluation, they analyzed the yearly changes in cultivated land area, construction land area, and ecological land area after the plan had been implemented. They found that year by year, arable land was lost and construction land expanded, which indicated that plan implementation did not achieve the desired effect.

Baer [14] believed that the type of evaluation criteria to be used depends on how the plan is viewed; for example, "a vision plan implies different criteria than for a blueprint plan, and a symbolic or expressive plan, different criteria than for an instrumental one". In fact, a plan can be both a blueprint and a vision, a development control tool, or a combination of various types. Therefore, the different types of plan implementation should be evaluated using different criteria and methods [13,14].

The polycentric system in master planning is both a blueprint and a vision. Planners hope that the public centers laid out in the blueprint can be built, and that the construction of public centers can play a practical role. Therefore, both criteria—materiality construction conformance and goal conformance—are indispensable in the evaluation.

## 2.2. Research of Big Data

In addition to determining the evaluation criteria, different types of plan implementation evaluation require suitable types of data that would reflect the reality in order to be compared with the plan. Previous evaluations have mainly focused on whether materiality construction conforms to the plan. Although several achievements have been made, better data can undoubtedly yield better results. The same is true for plan implementation evaluation of the polycentric system. When full sample built-environment data were not available to assess whether the public center had been constructed,

experience played a major role in identifying the public centers that had been built. Then, centers were classified according to the scale and the commercial forms obtained from surveys [24,25], which can be easily affected by subjective judgment. Particularly in the Chinese complex high-density built environment, it is difficult to assess the public center accurately and its scope by experience alone. In fact, researchers have preferred to define a public center as an area where a large number of people gather [2,26–28]; official surveys, such as the transportation survey and the census, can provide ideal data for the identification of these centers. The recent availability of point-of-interest (POI) data and the new technology of address resolution (through which an address can be converted to geographic coordinates) can easily locate a large number of facilities on the map. It is a type of location big data, which again makes it possible to identify public centers from the perspective of material space. Public centers can be identified through density and cluster analysis [29]. However, because there is a lack of building-area information, it is necessary to assume that the area of each point is the same and that the difference between identification result and reality cannot be tested. A more objective and accurate identification of built-up public centers requires full sample data containing the functions of the building and its floor areas.

Evaluation of whether the planning goals of the polycentric system have been achieved requires data from public activities of the residents. A large number of real sample data are required; traditional sampling or simulation results cannot meet the requirements. Because investigation through questionnaires can only gather samples regarding typical public centers, the results obtained via the simulation method can hardly verify whether public activities of the residents are in accordance with reality. That is to say, before the emergence of mobile-positioning big data (such as mobile phone data, bus card data, and floating car GPS data), data conditions for such research were not yet mature. It is mobile-positioning big data that has solved the data bottleneck. In recent years, using mobile-positioning big data, several research results regarding the spatio-temporal activities of residents have been obtained [30].

Some of these studies have already involved agglomeration characteristics of public activities in a city. Ratti et al. [31], Sevtsuk et al. [32] mapped the changes of human flow activities in urban spaces using mobile phone data, from which the agglomeration features of public activities were revealed. Roth et al. [33] conducted cluster analysis on passenger flow of London underground stations using metro card data; they found that people flow to multiple centers, which showed that London is a big city with a polycentric structure. Zhong et al. [34] analyzed passenger flow volumes of bus stations and subway stations with bus- and subway-card data. They found that in Singapore, travel distances and passenger flow volume grew with the improvement of bus and subway systems. The increase in passenger flow volume was mainly concentrated in new communities served by sub-centers, which proved that Singapore is transforming into a polycentric city structure. Arribas-Bel et al. [35] analyzed hotspots of urban activity during each hour in Amsterdam using mobile phone data; they found that during different periods, the stream of people that gathered in different areas presented different spatial structures.

Some of these studies involved the relationship between public activities and residential space. Becker et al. [36] compared the daily range of travel in three cities using call detail records; they found that people living in Los Angeles travelled longer distances on a typical day than people living in San Francisco, who, in turn, travelled longer distances than people living in Manhattan. Long et al. [37] mapped commuting trips from three typical residential communities to six main business zones and analyzed commuting patterns in Beijing with smart-card data. They found that these were highly consistent with the survey results. Zhou et al. [38] analyzed the relationship between the job–housing balance and self-containment of employment using mobile phone data; their study revealed that self-sufficiency was higher in the suburbs than in the center, which meant that the job–housing balance policy had a positive effect on self-containment of employment in the suburbs; however, it had a limited effect in the central area. Zhang et al. [39] also analyzed the job–housing balance using mobile

phone data and found that the commuting pattern of Shanghai was far from the extremes and that the relative balance of jobs with respect to housing was decent.

Although these studies were not involved in the evaluation of the results of plan implementation, they provided ideas for this study.

## 3. Methods and data

### 3.1. Research Area and Concept Definition

In this research, Hangzhou was selected as the example because it is a typical fast-growing city in China. The 2001 master planning has been steadily implemented as the basis for lower-level planning and subsequent policy-making. Nearly 20 years after planning implementation, it is time to evaluate the implementation results of the polycentric system that has been established during that time.

Hangzhou is located in the Yangtze River Delta, which is the fastest-growing economic region of China, and it is one of the central cities of the Yangtze River Delta city cluster. Since 2000, the economy of Hangzhou has been growing at a fast rate; this growth has been accompanied by a large influx of people and the rapid expansion of construction land. When master planning was developed in 2001, the urban construction land was only 201 km$^2$; it later increased to 538 km$^2$ when the new master planning started being drawn up in 2017. With the construction of peripheral areas, the single-center structure with daily public-service facilities that were concentrated in the old city has undergone a fundamental shift, at least in physical space. The urban development of Hangzhou is a typical example of Chinese cities changing from a single-center spatial structure limited to the old city, to a polycentric spatial structure, and represents one of the achievements in urban construction of fast development in China during the past 20 years. Moreover, great changes in the urban spatial structure can contribute to more thorough consideration of whether plan implementation of the polycentric system is successful.

In this research, the polycentric system refers to the internal multiple-center system of the city, corresponding to the public-center system planning in city master planning. The research area is the core built-up area of Hangzhou, within the city highway, covering an area of 955 km$^2$ (see Figure 1). Within this scope, urban construction land stretches into patches, which are significantly different from the built environment outside the scope. The 2001 master planning deployed three principal centers, three secondary centers, and ten sub-secondary centers within its scope.

In this research, public activity refers to the activities conducted in public spaces. According to the four major urban functions presented in the Athens Charter, employment and recreation are the two most important public activities. In the present work, public activity means employment activities and recreational activities.

In this study, the term public center refers to areas that provide space for public activities of mass agglomeration of the population that tend to have large numbers of public-service buildings, including cultural, commercial, and commercial buildings (hospitals and schools, which are rarely related to employment and recreation, have been excluded). The land use types of these buildings correspond to the cultural facilities land, commercial land, business land, entertainment and sport land, and commercial- and business-integrated land in the classification of land use in China. Land for serviced apartments, furniture stores, and wholesale markets have been excluded because serviced apartments are often used as residence and services provided by furniture stores and wholesale markets are not public services needed on a daily basis. These areas will be referred to as public-service land in the following sections.

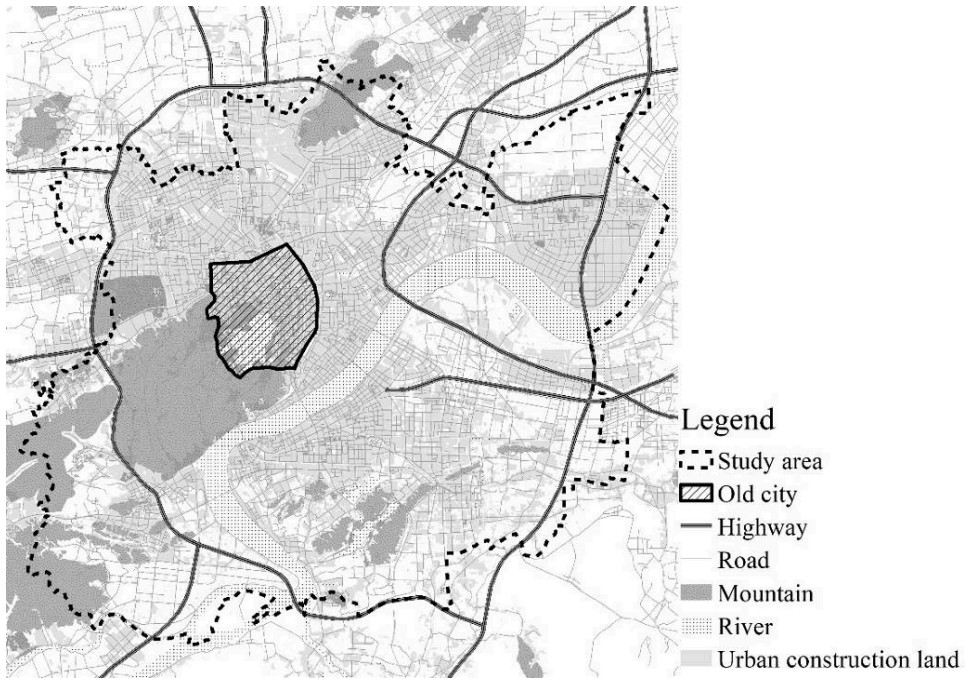

**Figure 1.** Study area.

*3.2. Evaluation Framework and Criteria*

Figure 2 presents the evaluation framework of this research. The evaluation criteria will be presented in the following subsections.

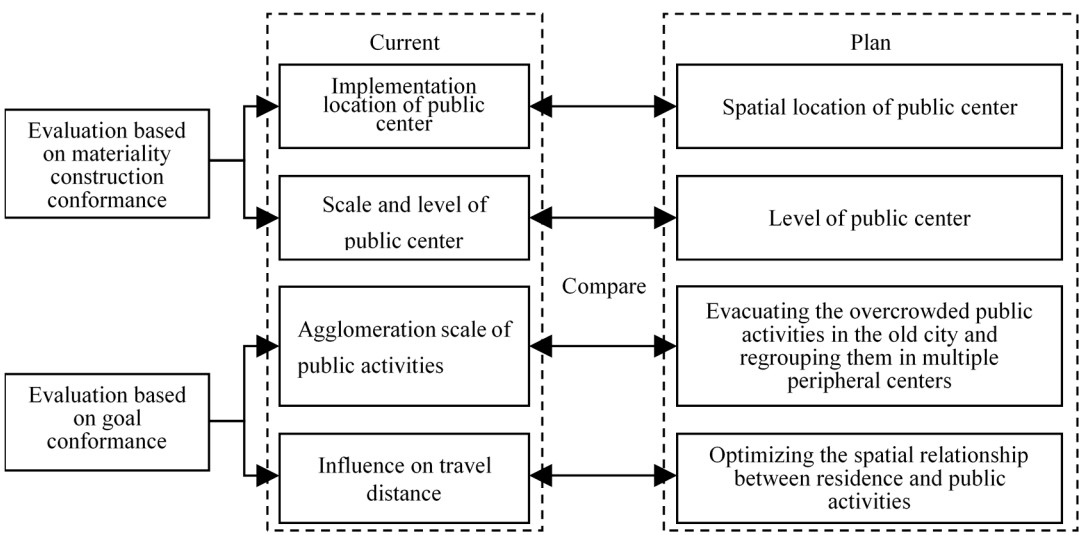

**Figure 2.** Evaluation framework.

3.2.1. Evaluation Based on Materiality Construction Conformance

The evaluation based on materiality construction conformance is the use of full sample built-environment data for the analysis of implementation location, scale, and level of current public centers, and for the comparison of conformance degree between the aforementioned built-up centers and the original plan.

1.　Evaluation criteria of the implementation location: public center system planning is structural and the public center identified in planning drawing or text is only an intention; thus, the actual location of the public center is allowed to deviate from the plan. As long as it is close to the planned location, the public center can be considered to have been implemented according to the planned location.

2.　Evaluation criteria of the scale and level: the planning specifies only the level of the public center, but not the construction scale of each level. As long as the actual construction scale of the principal center is obviously larger than that of secondary center, and the secondary center is obviously larger than the sub-secondary center, the public center can be considered to have been implemented according to the planned level.

### 3.2.2. Evaluation Based on Goal Conformance

The evaluation based on goal conformance involved the use of mobile-positioning big data to analyze the agglomeration scale of public activities and their influence on the travel distance for public activity of built-up public centers, as well as to compare and confirm whether they are inconsistent with the original plan.

1.　Evaluation criteria for the public activity agglomeration scale: if the built-up public center is able to gather a larger scale of public activity than the non-public center area—the larger the scale of construction is, the larger the scale of public activity—the public center is considered to have achieved the goal of dispersing the overcrowded public activities from the old city and of re-gathering them in multiple peripheral centers.

2.　Evaluation of the influence on travel distance for public activities: theoretically, the successfully implemented polycentric system should help reduce travel distance for public events, compared with the single-center system of the same scale. However, the single-center system of the same scale in the same city does not exist; thus, a comparison cannot be made. In previous studies, alternative methods have been used for comparison, such as comparing the travel distance difference between a polycentric city and a single-center city or comparing the travel distance difference between the principal-center area and the secondary-center area of the same city [3–5,40,41]. In the present research, the alternative method was employed as well. Thus, in this work, the difference in current travel distances between people living near the public center and that of people living away from it will be compared. That is to say, if a built-up public center is able to provide public services to nearby blocks, the closer residents live to the public center, the shorter the distance from the public activities. If this goal could be achieved, the public-center construction could be considered as having achieved the goal of optimizing the spatial relationship between residence and public activities.

### 3.2.3. Conclusions of Conformance Evaluation

The above four evaluation criteria are independent of one another. The result that planners want to obtain the most is that both materiality construction conformance and goal conformance are high; however, this rarely occurs in reality. Its opposite situation is that both materiality construction conformance and goal conformance are low, indicating that plan implementation is almost a complete failure, which is the least desired result. In reality, there are other possibilities. For example, materiality construction conformance is high, but goal conformance is very low. This may be caused by one possible extreme situation, in which the public center has been built as planned but the utilization rate of public buildings in the peripheral center is low. Conversely, the utilization rate could be high, but public activities gathered in peripheral centers are attributed to the old city and public activities gathered in the old city are attributed to peripheral areas. Such a result would not be considered a complete failure; however, it is clearly not a success either. The other possibility is that materiality construction conformance is low, but goal conformance is high. In this case, the extreme situation may

be that the public center has not been built at the designated location but at a different peripheral area. The construction result seems to deviate from plan; however, if the public center can gather public activities and reduce the travel distance of the residents as expected, it is still a successful implementation. Therefore, the evaluation of the plan implementation result needs to analyze the situation in detail; moreover, the specific content of failure should be discussed to provide reference for subsequent planning.

*3.3. Data*

In this research, two types of big data were used, namely the full sample built-environment data and the mobile-positioning big data. The full sample built-environment data refers to the present situation of land use (provided by the government, including land use property information for each plot) and buildings (crawled from the online map using crawler software, including the area and floor-number information of each building). The data were collected at the end of 2016.

Mobile-positioning big data refer to the destination (workplace and recreation area) and the departure location (residence) of the public activities of residents. The data were collected from mobile phone signaling data provided by China Mobile in the Zhejiang Province for 30 continuous days in April, 2017. The original data recorded the individual movement trajectory of each mobile phone user. The locations were identified using base stations. Through the judgment of the base stations to be connected in nights and daytime of working days, residence and employment locations were identified; by judging the small area other than working and living locations where they stay continuously during their day off, recreation destinations were confirmed. These data processing methods referred to the research of Ahas [42], which has been relatively mature at present. In this manner, working–living locations of 2.34 million local permanent residents were identified and recreation–living locations of local permanent residents with a person-time of 38.25 million were identified. The travel distance refers to the linear distances between the residence and working location, as well as between the recreational area and the residence.

These data are equivalent to a large-sample survey on public activities of residents. And the residence locations, employment locations and recreation destinations can be located by coordinates of base stations. In the core built-up area, there is about one base station for every 9 hm$^2$, the positioning accuracy is much higher than the official data.

These data were verified for accuracy. At the 1% significance level, the correlation coefficient between the residence distribution and official statistics was 0.93, while this result of the employment distribution is 0.52, which are strong correlation. There are no official statistics for recreation destinations to check, but the daily distribution results are highly similar, suggesting that the data should be accurate. These test results show that the data can reflect the real public activity characteristics of residents.

## 4. Results

*4.1. Evaluation Based on Materiality Construction Conformance*

Considering a block as a statistical space unit, in this study, we will summarize the acreage of constructible land (excluding green spaces, squares, roads, water area, etc.) and the gross floor area of various types of land of each block; furthermore, public-service land and other lands will be distinguished. Buildings on public-service land are public-service buildings, whereas buildings on other lands are non-public service buildings. Each block was assessed, to identify whether it belonged to a public center according to the following criteria (all criteria must be met simultaneously):

1.   The average floor area ratio of the public-service land is ≥1;
2.   The proportion of the public-service land is ≥20%;
3.   The proportion of the gross floor area of public-service buildings is ≥50%.

In total, 69 blocks exist that meet the criteria. By combining adjacent or closely spaced blocks, 15 public centers were identified, among which the Huanglong, and the Yan'an Road were located in the old city, corresponding to the earliest employment center and business center of Hangzhou, respectively (see Figure 3).

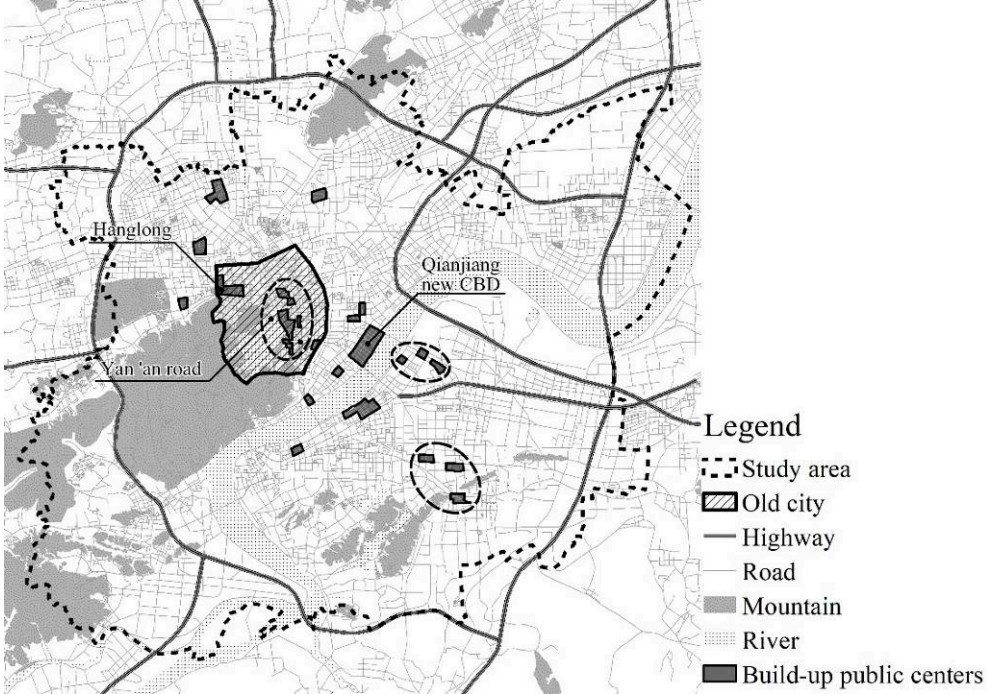

**Figure 3.** The built-up public centers. Note: The blocks enclosed within the dotted line are merged into a common center.

The comparison between locations of built-up public centers and original planned public centers shows that of the original planned 11 centers (professional centers excluded), 6 have been built (see Figure 4 and Table 1). Moreover, two professional sub-secondary centers have been built into comprehensive centers (the professional center in the planning refers to professional markets, and exhibitions, which provide non-daily public services, not in line with the definition of a daily public center). Given that the original planning period was from 2001 to 2020, there are four centers under construction; two have not been built yet but they still have a construction space. It is estimated that by the end of the planning period, 9 of the 11 public-service centers originally planned to provide daily public services will have been completed, along with two original planned professional centers transformed into comprehensive centers and seven additional public centers that have been built in the periphery. Thus, the implementation rate of the polycentric system is relatively high. However, in terms of the spatial distribution of the built-up public centers, three outermost public centers will not be completed, thus indicating that the spatial composition of the polycentric system is not perfect.

Among the built-up public centers, the largest is the principal center, namely the Qianjiang new central business district (CBD), with a public-service floor area of over 4.84 million m$^2$; the second largest is the Yan'an Road, with an area of over 2.490 million m$^2$. Both areas of the two secondary centers reach 1.6 million m$^2$. Although the scale of the sub-secondary center is smaller than that of secondary center, all sub-secondary centers exceed 1,000,000 m$^2$. The seven unplanned public centers are slightly smaller, ranging from 200,000 m$^2$ to 900,000 m$^2$. The scale of each of the principal, secondary, and sub-secondary centers decrease in gradient, with a primacy ratio reaching 1.9. Therefore, the current scale and level can be considered as being consistent with the plan in general.

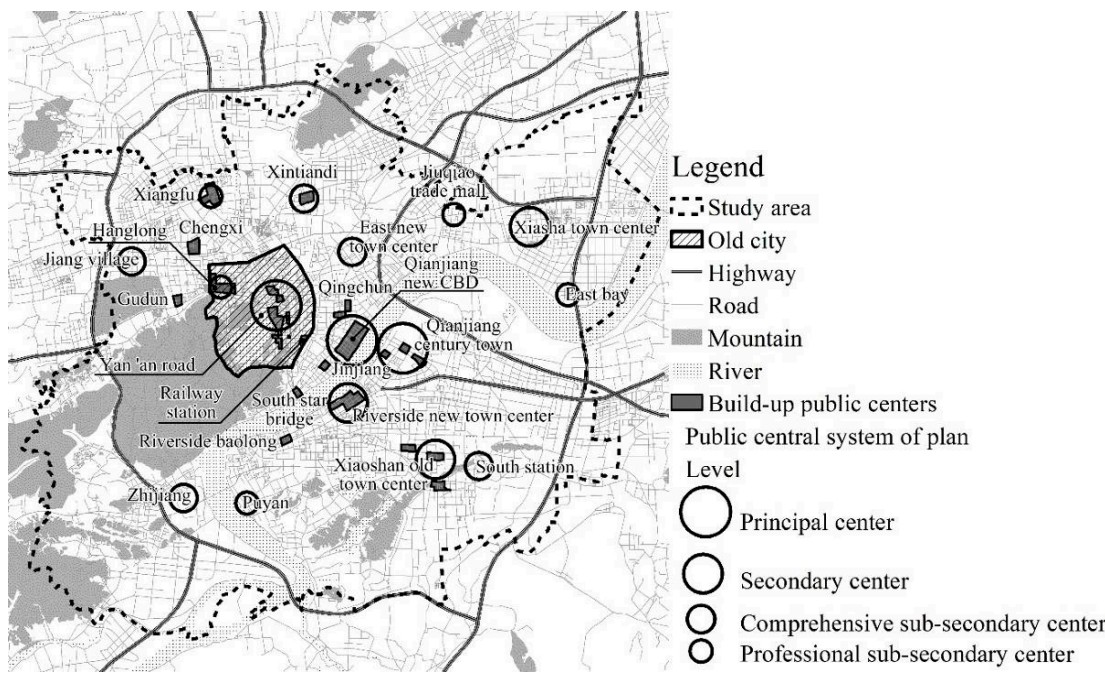

**Figure 4.** Conformance comparison between public centers' reality and plan.

**Table 1.** Comparison of public centers of implementation and plan.

| Name | Planned Level | Planned Function Type | Build-Up Floor Area of Public-Service Buildings (million m$^2$) | Remark |
|---|---|---|---|---|
| Yan'an road | Principal | Tourism and business | 2.49 | Have been built |
| Qianjiang new CBD | Principal | Business and commerce | 4.84 | Have been built |
| Qianjiang century town | Principal | Business | 0.85 | Under construction |
| Xiaoshan old town center | Secondary | Comprehensive | 1.6 | Have been built |
| Riverside new town center | Secondary | Comprehensive | 1.6 | Have been built |
| Xiasha town center | Secondary | Comprehensive | | Under construction |
| Xintiandi | Sub-secondary | Comprehensive | 1.11 | Have been built |
| East new town center | Sub-secondary | Comprehensive | — | Under construction |
| Jiang village | Sub-secondary | Comprehensive | — | Under construction |
| Zhijiang | Sub-secondary | Comprehensive | — | No construction |
| South station | Sub-secondary | Comprehensive | — | No construction |
| Xiangfu | Sub-secondary | Profession | 1.03 | A comprehensive center has been built |
| Huanglong | Sub-secondary | Profession | 1.26 | A comprehensive center has been built |
| Jiuqiao trade mall | Sub-secondary | Profession | — | — |
| Puyan | Sub-secondary | Profession | — | — |
| East bay | Sub-secondary | Profession | — | — |
| Qingchun | Unplanned | — | 0.9 | Have been built |
| Chengxi | Unplanned | — | 0.73 | Have been built |
| Jinjiang | Unplanned | — | 0.43 | Have been built |
| South star bridge | Unplanned | — | 0.41 | Have been built |
| Riverside baolong | Unplanned | — | 0.32 | Have been built |
| Railway station | Unplanned | — | 0.29 | Have been built |
| Gudun | Unplanned | — | 0.2 | Have been built |

To summarize, the results of the evaluation based on materiality construction conformance showed that the plan implementation of the polycentric system deviated from the original plan to a certain extent. However, most of the principal and secondary centers were built as per the plan, and the unplanned built-up public centers compensated for the influence of the unbuilt public centers on the construction of a polycentric system (to a certain extent). The successful construction of the Qianjiang

new central business distric (CBD)—the newly planned principal urban center—achieved a shift of the spatial structure from a single main center to two main and multiple secondary centers. In future, with the construction and implementation of the remaining public centers, the polycentric system will be improved further. Overall, the materiality construction of the polycentric system was successful.

*4.2. Evaluation Based on Goal Conformance*

4.2.1. Effect of Dispersing and Re-Gathering Public Activities

First, differences between the agglomeration levels of the public-center and the non-public center areas were compared. As can be seen from Figure 5, the scale of public activity agglomerated per unit area of the public center was significantly higher than that of non-public center areas; the farther from the public center, the lower the agglomeration level. As the distance from the public center becomes larger, the agglomeration of employment and recreational activities per unit area drops dramatically in scale, with a dropping rate as high as 50% within the range of a block (0.5 km). The pubic centers attribute for only 1.3% of the studied area; however, they account for 7.9% and 10.9% of agglomerated employment activity and recreational activity. The public centers plus a nearby block of each center takes up only 6.4% of the studied area; nevertheless, the agglomerate employment activity and recreational activity is as high as 20.2% and 23.7%, respectively.

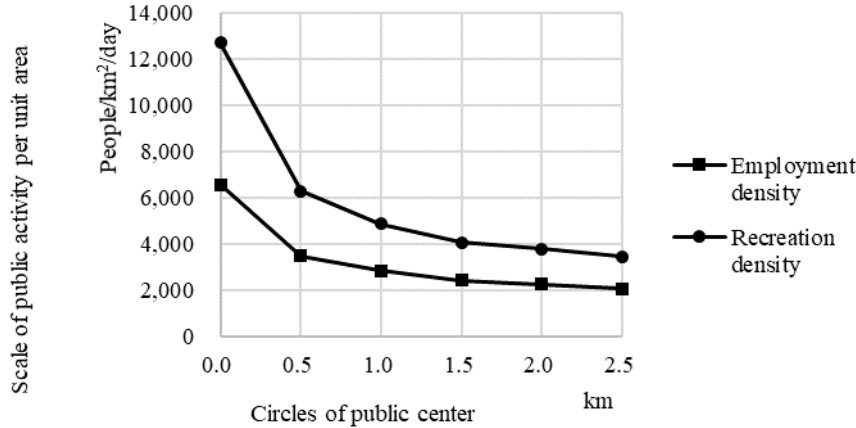

**Figure 5.** Scale of public activity agglomerated per unit area of public center circle. Note: the public centers were divided into different circles with a discontinuous boundary of 0.5 km; then, scale of public activity per unit area of each circle was calculated.

Next, the relationship between the construction scale and public activity agglomeration scale of each public center was analyzed. It can be seen from Figure 6 that, in general, the public centers conform to the law that the larger the scale of a construction, the larger the scale of the public activity (the fitted curve passed the significance testing, with a value of 1%). With the increase in the construction scale, the marginal utility of the public center in agglomerating public activity diminishes progressively. Except for the Huanglong and Yan'an Road in the old city, the public activity of the remaining 13 public centers slightly deviates from the fitting curve (under the significant level of 1%, the deviations from the fitting curve show no abnormal value). The employment scale in Huanglong is obviously higher than the theoretical employment agglomeration level for the construction scale of 1.26 million m$^2$ (under a significant level of 1%, the employment scale deviates from the fitting curve, presenting an abnormal value). According to the fitting curve, the employment scale in Huanglong should be only 6920 people per day; however, the actual number reaches 19,232 people per day, with a primacy ratio as high as 1.7 (see Table 2). For the Yan'an Road, its recreation scale is significantly higher than the theoretical agglomeration level of the recreational activity for a construction scale of 2.49 million m$^2$ (under a significant level of 1%, the recreation scale deviates from the fitting curve, showing an

abnormal value). According to the fitting curve, the recreation scale of the Yan'an Road should be only 21,285 people per day; however, the actual number reaches 53,385 people per day, with a primacy ratio as high as 2.4 (see Table 2).

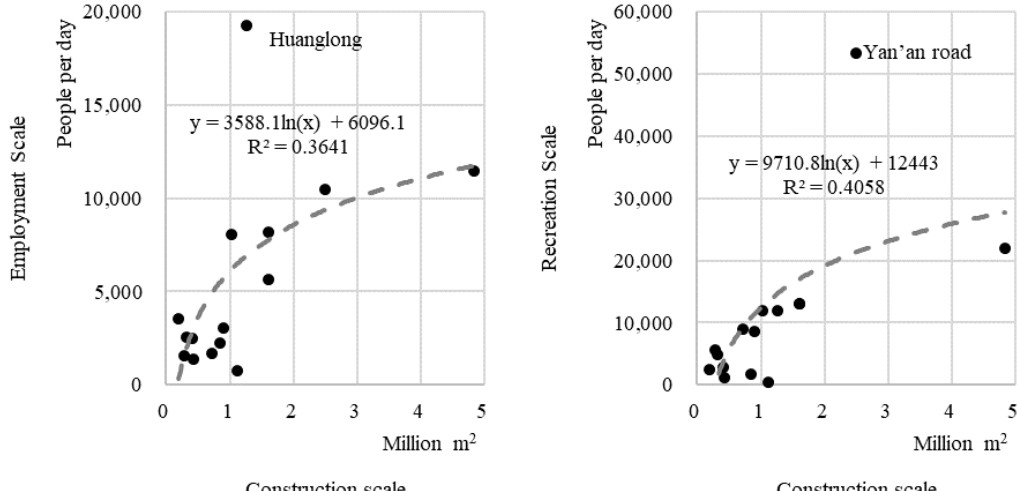

(**a**) Employment scale–construction scale fitting curve.　　(**b**) Recreation scale–construction scale fitting curve

**Figure 6.** Fitting curve of public activity scale and construction scale for each public center. **Note:** 1. The scale of public activity represents only the number of people identified from mobile phone signaling data but not the true value (the same below). 2. The recognition rates between employment activity and recreational activity obtained from mobile phone signaling data are different; thus, they are of no horizontal comparability.

**Table 2.** Standard residual error and theoretical public activity scale calculated from the fitting curve.

| Name | Construction Scale (million m²) | Employment | | | Recreation | | |
|---|---|---|---|---|---|---|---|
| | | Standard Residual Error | Actual Employment Scale | Theoretical Employment Scale Calculated from the Fitting Curve | Standard Residual Error | Actual Recreation Scale | Theoretical Recreation Scale Calculated from the Fitting Curve |
| Yan'an road | 2.49 | 0.26 | 10,452 | 9363 | 3.05 | 53,385 | 21,285 |
| Qianjiang new CBD | 4.84 | −0.06 | 11,479 | 11,751 | −0.55 | 21,918 | 27,748 |
| Qianjiang century town | 0.85 | −0.76 | 2268 | 5494 | −0.86 | 1727 | 10,812 |
| Xiaoshan old town center | 1.60 | −0.50 | 5672 | 7783 | −0.37 | 13,075 | 17,007 |
| Riverside new town center | 1.61 | 0.09 | 8161 | 7798 | −0.37 | 13,125 | 17,049 |
| Xintiandi | 1.11 | −1.34 | 770 | 6483 | −1.23 | 499 | 13,491 |
| Xiangfu | 1.03 | 0.45 | 8086 | 6190 | −0.08 | 11,855 | 12,696 |
| Huanglong | 1.26 | 2.89 | 19,232 | 6920 | −0.26 | 11,918 | 14,673 |
| Qingchun | 0.90 | −0.62 | 3081 | 5726 | −0.27 | 8574 | 11,441 |
| Chengxi | 0.73 | −0.76 | 1700 | 4948 | −0.03 | 8997 | 9336 |
| Jinjiang | 0.43 | −0.40 | 1386 | 3097 | −0.29 | 1253 | 4327 |
| South star bridge | 0.41 | −0.09 | 2493 | 2879 | −0.08 | 2892 | 3735 |
| Riverside baolong | 0.32 | 0.13 | 2527 | 1,973 | 0.34 | 4887 | 1285 |
| Railway station | 0.29 | −0.02 | 1573 | 1,664 | 0.49 | 5580 | 447 |
| Gudun | 0.20 | 0.75 | 3534 | 346 | 0.54 | 2531 | −3,119 |

Note: Standard residual error greater than 1.96 or less than −1.96 indicate under significant level of 1%, the public activity scale deviates from fitting curve significantly.

The above indicates that built-up public centers have played a role in gathering public activities. Moreover, the pubic activity agglomerate scales of peripheral public centers are highly consistent with their construction scale, thus playing the role of guiding public activities to agglomerate in peripheral areas. However, compared with the two public centers in the old city, the agglomeration level of other

public centers is still weak and has failed to moderate excessive concentration of public activities in the old city.

4.2.2. Effect of Optimizing the Spatial Relationship between Residence and Public Activity

In this study, we analyzed whether the polycentric system has achieved the effect of optimizing the spatial relationship between residence and public activity by using the data on travel distance of residents for public activities. The ideal pattern of the travel distance distribution is that the area closer to the public center would enjoy a shorter travel distance (see Figure 7).

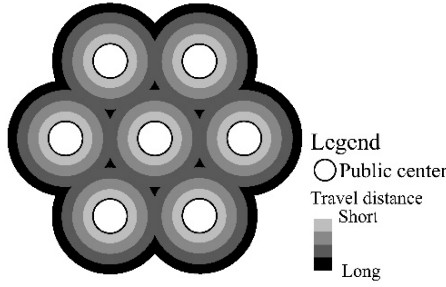

**Figure 7.** The ideal pattern of travel distance distribution.

Based on the average travel distance required for residents to participate in public activities, as summarized in the base station data, the travel distance for the public-activities distribution diagram was obtained via the Kriging interpolation analysis (see Figure 8). It can be seen from the diagram that the travel distance of residents living around public centers is not shorter than of those living in peripheral areas. For employment travel distance, although in certain public centers residents living around them do have a shorter travel distance, travel distance does not increase significantly with the increase in distance from the public center. For recreational travel distance, areas with the longest travel distance are distributed in locations near public centers, which differs even more from the ideal pattern.

To prove these observations further, the surrounding area of a public center was divided into different circles with 1 km as a grade; the average travel distance for the public activities of each circle was measured. As can be seen from Figure 9 that travel distance does not increase circle by circle as expected; instead, it presents a wavy and irregular feature of first descending, then rising, descending again, and finally rising again. The fluctuation ranges are not large, at only 607 m and 765 m for the employment and recreational travel distances among circles within 15 km from the public center, respectively.

The above indicates that built-up public centers have had little effect on encouraging people to visit the nearest blocks for daily public activities and have no obvious effect on optimizing the spatial relationship between residence and public activities.

Thus, the results of the evaluation based on goal conformance show that built-up public centers have played a role in gathering public activities in peripheral areas; however, they played a limited role in moderating excessive agglomeration of public activities in the old city. In addition, built-up public centers failed to encourage residents to visit nearby blocks; hence, the spatial relationship between residence and public activities needs to be further optimized. In general, the polycentric system has not yet achieved the planning goals.

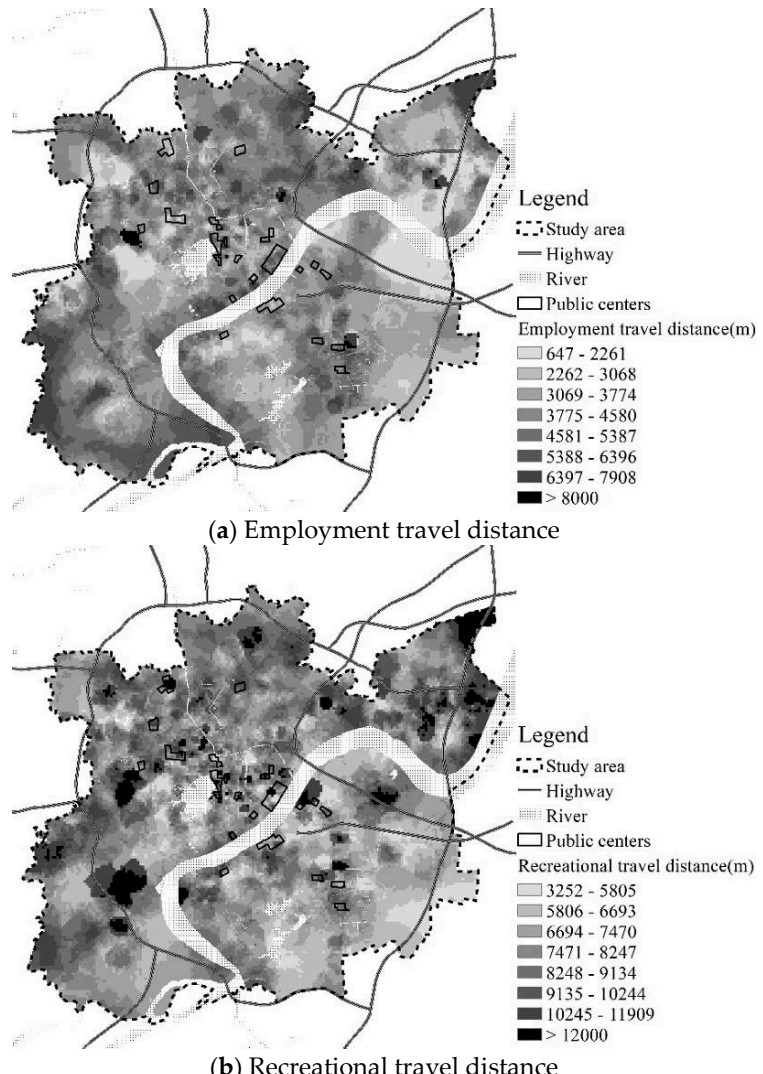

(**a**) Employment travel distance

(**b**) Recreational travel distance

**Figure 8.** Travel distance for public activities distribution diagram. Note: for display convenience, the travel distance distribution was divided into seven grades with natural discontinuity classification after eliminating the abnormal values.

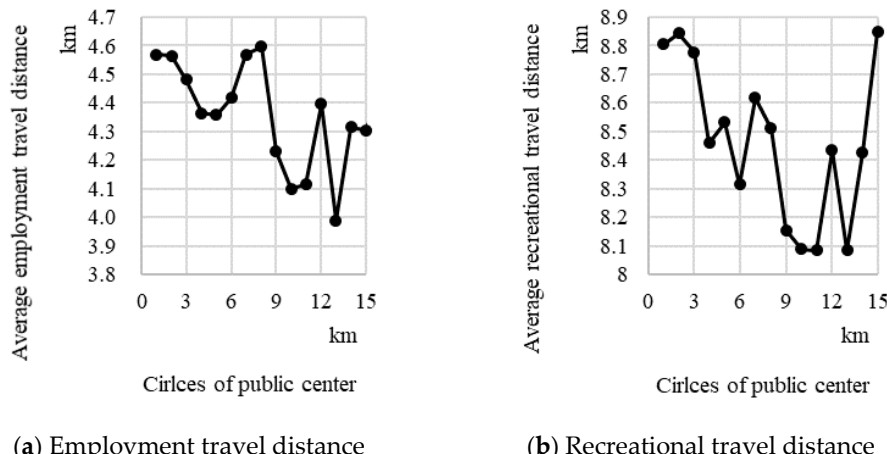

(**a**) Employment travel distance

(**b**) Recreational travel distance

**Figure 9.** Travel distance for public activity distribution statistics of circles.

## 5. Discussion

The results of evaluations based on materiality construction conformance and on goal conformance were nearly opposite. Nevertheless, materiality construction has not been a complete success, nor were the planning goals entirely unmet. In this section, the reasons for the results of the aforementioned plan implementation will be discussed.

Materiality construction of the Hangzhou polycentric system has been implemented relatively successfully. The reasons are the following. First, Hangzhou has adopted an active urban construction policy since 2008. The 2008 Opinions on Accelerating the Development of Building Economy put forward that "220 new commercial buildings will put into use with overall floorage reaching over 16 million $m^2$"; by 2010, 245 buildings will be planned. The 2009 Opinions on Building a Networked Metropolis and the Construction of New Cities and Urban Complexes proposed "to build 20 new cities and 100 urban complexes". In addition, the 2011 Opinions on Promoting the Construction of "Shopping Paradise and Food Capital" suggested to "focus on developing the construction of 100 key business projects". These projects were implemented through the 11th Five-Year Plan (2006–2010) and the 12th Five-Year Plan (2011–2015) of the national economic and social development plan for major construction projects in Hangzhou and other administrative regions. The commercial buildings, complexes, and key business projects involved in the above policies are the materiality subjects that constitute the public-center system. Under a strong policy impetus, the public centers planned in the 2011 master planning were implemented. In particular, the Qianjiang new CBD has been built into a new (and the largest) public center in less than 20 years, which prompted the public-center system to shift from a single-center system to a system of two main centers and multiple sub-centers. Second, Hangzhou has been exploring and implementing land reform policy for urban regeneration since 2001. For the original village areas incorporated into the urban scope, the original land owners are encouraged to cooperate with developers to redevelop the land according to planned use. Developers can use construction funds in exchange for the free use of up to 50% of the building area for 30 years, without the need to spend a large sum of money to buy land. The aforementioned would greatly reduce the cost of land development and would improve the enthusiasm of land redevelopment in peripheral areas. Much of the redeveloped land has been used to build commercial buildings and complexes, thus facilitating the formation of public centers in the periphery. Under the influence of the above policy factors, Hangzhou has completed the material transformation of the public center from a single main center to two main and multiple secondary centers.

However, the planning goals of the Hangzhou polycentric system have been implemented unsatisfactorily. The reasons are as follows. First, the center system has a path dependence on the original public center in the process of evolution. Even though a new public center has been built on the periphery, it is influenced by the law of the agglomeration economy; thus, it is difficult for enterprises to escape the original center and re-gather at the new center. In this case, peripheral public centers only offer limited jobs and business services, hence being unable to moderate the overcrowded public activities in the old city. In other words, at present, the old city still has the capacity to agglomerate public services; the negative effect of agglomeration is not enough to lead enterprises to seek new development space in peripheral areas. Second, the spatial optimization of public activities and the construction of the urban material space are mutually coordinated processes, in that residents consider factors belonging in both of them, such as housing cost, transport cost, housing quality, and public-facility service level. This is the result of the continuous optimization of the local governments of the spatial allocation of residential and public-service settings, which would require a significant amount of time. However, the mega-cities of China, such as Hangzhou, are expanding too fast, while the construction of the polycentric system is mainly driven by planning and policies instead of being a natural process. Upon the completion of new public centers in the periphery, residents have not yet adjusted well in the spatial relationship between the living and the public activities.

## 6. Conclusions

The polycentric system in master planning is a structural plan, which is different from a physical blueprint. Its evaluation focuses on answering whether the vision of the polycentric system is realized. The polycentric system in the material space may not necessarily guide public activities to gather in multiple centers; meanwhile, the agglomeration of public activities is not necessarily the result of the optimized spatial relationship between residence and public activity. As described in the introductory section, if public activities are dispersed and re-gathered but residents of peripheral areas still need to travel to the old city for daily activities, and residents of the old city to peripheral centers (in other words, the polycentric system seems to have been established) then "big city malaise" is exacerbated. Therefore, the evaluation of plan implementation of the polycentric system requires evaluation of materiality construction, as well as the attention to planning goals.

Comparing the reality with the plan is a basic means to assess implementation results. The difficulty in this is quantifying reality. The emergence of full sample built-environment data and mobile-positioning big data simplifies this process. Without relying on assumptions and models, reality can be modelled through spatial statistics and data analysis and the evaluation of planning goals can be realized; in the past, this was unachievable. Ultimately, the evaluation can be improved objectively and accurately.

In this study, full sample built-environment data and the mobile-positioning big data were used to evaluate the plan implementation results of the polycentric system. First, we evaluated whether plan implementation is successful based on the outcome of materiality construction using the data of the built environment. Using the case of Hangzhou, it was found that in the Hangzhou core built-up area, 15 city-level public centers exist, among which 8 were public centers that had been identified in the original plan and 7 had not been included in the original plan. Although the current situation deviates from the plan, the unplanned built-up public centers compensate for the influence of unbuilt public centers on the construction of the polycentric system to a certain extent. In terms of material space, the plan implementation of the polycentric system has successfully guided the transformation of urban spatial structure from a single main center to two main and multiple secondary centers. Second, the evaluation was conducted based on the outcome of planning goals using mobile-positioning big data. It was found that although peripheral public centers could gather public activities matching their construction scales, the public activity agglomerate level of the two public centers in the old city is still significantly higher. For residents living near a public center, their travel distance is not shorter than those living far away from the public-center area. Regarding the implementation effect, the plan implementation has successfully realized the goals of gathering public activities in peripheral public centers. However, it failed to relieve overcrowded public activities of public centers in the old city, and did not achieve the optimized spatial relationship between residence and public activity. Finally, the factors influencing the plan implementation results of the polycentric system were discussed. A strong plan execution capacity on the part of government and positive land redevelopment policies can guarantee that the materiality construction of the polycentric system will be carried out according to the blueprint to form a polycentric system in material space within a short period. However, as the city development has its own law, the goals of a polycentric system are difficult to be achieved within a short period.

**Author Contributions:** L.D. conceived the idea and analyzed it; L.D. and C.S. wrote the manuscript; X.N. participated in the acquisition and processing of data. All authors have read and agreed to the published version of the manuscript

**Funding:** This research was funded by National Natural Science Foundation of China, No. 51808495 and 51808392.

**Conflicts of Interest:** The authors declare no conflict of interest.

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
