# Peer review of "Evaluation of Plan Implementation in the Fast-Growing Chinese Mega-City: A Case of a Polycentric System in Hangzhou Core Built-Up Area"

_sustainability, doi:10.3390/su12051723_

Round 1

Reviewer 1 Report

It is a really interesting and well organised paper. It results very useful to a wide science community which deals with city regeneration and adaptation policies. The paper provides a clear picture of evaluating plan implementations: methods and data are well and clearly reported. I suggest it for publication after having properly solved the error reported at line 391.

Author Response

Point 1: It is a really interesting and well organised paper. It results very useful to a wide science community which deals with city regeneration and adaptation policies. The paper provides a clear picture of evaluating plan implementations: methods and data are well and clearly reported. I suggest it for publication after having properly solved the error reported at line 391.

Response 1: There was a table reference error at line 391, the revised version is at line 399. It has been corrected, and we added the correct reference of table 2 at line 406 and line 411.

Reviewer 2 Report

This was an interesting study to understand the polycentric system in one Chinese mega-city. Overall, the paper looks good and I just have a few comments as follow:

1. It is not clear for me how the mobile phone data was assigned to the public center. Is there any data cleaning process? Sometimes the trajectory data could be biased if not well-processed. This paper below might be of interest for mobile phone data processing.

Shen Y., Karimi K., Law S., and Zhong C. (2019), Physical co-presence intensity: Measuring dynamic face-to-face interaction potential in public space using social media check-in records, PloS one, 14(2): e0212004.

2. The decision to use Concentric zone theory (Figure 7) needs to be better motivated. Why was not other models considered, such as central place theory or multiple nuclei model?

Author Response

Point 1: It is not clear for me how the mobile phone data was assigned to the public center. Is there any data cleaning process? Sometimes the trajectory data could be biased if not well-processed. This paper below might be of interest for mobile phone data processing.

Shen Y., Karimi K., Law S., and Zhong C. (2019), Physical co-presence intensity: Measuring dynamic face-to-face interaction potential in public space using social media check-in records, PloS one, 14(2): e0212004.

Response 1: The distribution of the mobile phone data to the public center is a question of how is the data located to the space. It had been proposed at line 311 that the trajectory data are located by base stations, but it did not explained how to locate the processed data. So we add this at lines 320-322.

The data processing process had been explained at lines 311-314. The revised version has added a reference related to mobile phone data processing, and added the data testing at lines 320-329.

Point 2: The decision to use Concentric zone theory (Figure 7) needs to be better motivated. Why was not other models considered, such as central place theory or multiple nuclei model?

Response 2: Figure 7 is not Concentric zone theory, but an ideal pattern of travel distance distribution around public centers. This was drawn according to line 276-278: “if a built-up public center is able to provide public services to nearby blocks, the closer residents live to the public center, the shorter the distance from the public activities”. However, the original figure only showed one public center, the distribution of travel distance around multiple public centers may not be easy to understand. Therefore, we modified it to show the travel distance distribution around multiple public centers. The travel distance distribution around one or more public centers can be deduced from this figure.